# Diarrhoea among Children Aged 5 Years and Microbial Drinking Water Quality Compliance: Trends Analysis Study in South Africa (2008–2018)

**DOI:** 10.3390/ijerph20010598

**Published:** 2022-12-29

**Authors:** Mohora Feida Malebatja, Mpata Mathildah Mokgatle

**Affiliations:** 1Environmental and Occupational Health Department, Health Sciences Faculty, School of Health Systems and Public Health, University of Pretoria, Private Bag X 20, Hatfield, Pretoria 0028, South Africa; 2Environmental and Occupational Health Division, Public Health Department, School of Healthcare Sciences, Sefako Makgatho Health Sciences University, Pretoria Ga-Rankuwa, Pretoria 0208, South Africa

**Keywords:** diarrhoea, water quality, children, sanitation, microbiological water quality, child mortality and morbidity rates, compliance status, South Africa

## Abstract

In developing countries such as South Africa, diarrhoeal diseases are reported to be linked to inadequate drinking water quality, sanitation, and hygiene behaviours. The consumption of microbiologically contaminated drinking water has been reported to cause diarrhoea, mortality, and morbidity in children under the age of five years. This study evaluated the microbiological pathogens detected in municipal drinking water and diarrhoea trends for children under 5 years of age in South Africa between 2008 and 2018. A trends analysis study was conducted using secondary data on diarrhoea for children under the age of five years and microbial drinking water quality compliance. A negative correlation was found between the occurrence of microbial pathogens in water, morbidity, and mortality rates of children under the age of 5 years in South Africa. As compliance status improved, the mortality rate among children under 5 years old decreased by 31% over the study period. A conclusion can thus be drawn that the microbiological pathogens detected in drinking water at levels complying with SANS 241:2015 Edition 2 standards and diarrhoea incidences were not the primary cause of the mortality of children under 5 years old in South Africa between 2008 and 2018.

## 1. Introduction

Water quality refers to the physical, chemical, aesthetic, and microbiological properties of water that makes it fit for use [1,2]. The microbiological quality of water refers to the biological agents and microorganisms present in water, such as viruses, bacteria, protozoa, parasites, and fungi [1,2]. The consumption of poor microbiological water quality can cause gastrointestinal infections such as diarrhoea, dysentery, typhoid fever, shigellosis, cholera, human enteritis, and cryptosporidiosis, particularly in children under the age of five years [3,4,5].

The microbiological quality of water is of great public health importance due to the high mortality rates of children under the age of five years linked to the consumption of poor-quality water [6]. Sustainable Development Goal 3.2 is aimed at reducing child mortality to at least 2.5% by 2030 in all countries [7]. Studies have emphasised that diarrhoeal diseases are the second largest cause of mortalities amongst children under the age of 5 years and are linked with exposure to contaminated drinking water [8]. Therefore Sustainable Development Goal 3.2 deals with factors that could potentially have an impact on child mortality and morbidity resulting from diarrhoeal diseases [7]. The Sustainable Development Goal 6 of the 2030 target for clean water and sanitation is to achieve universal and equitable access to adequate and affordable drinking water that is safe for human consumption for all [6].

In order to avoid potential health risks, such as outbreaks of infectious diseases associated with extremely high concentrations of pathogenic microorganisms found in potable water, the World Health Organization (WHO) issued drinking water quality guidelines. The South African National Standards (SANS) for drinking water quality 241:2015 Edition 2 gives the national standards for potable water for the South African population that should be adhered to [1,2].

According to the South African National Standards (SANS) 241-1:2015 Edition 2 for drinking water, the microbiological determinants of drinking water quality include *Escherichia coli, Faecal coliforms*, *Cryptosporidium species*, *Giardia species*, *Total coliforms*, and *Heterotrophic plate count* [1,2]. The presence of these microbiological pathogens in an amount that exceeds the established standards and guidelines in drinking water serves as an acute health risk to the public [9,10,11].

The perpetual ingestion of microbiologically contaminated water by consumers could lead to episodes of waterborne and water-related diseases, epidemics, and pandemics [3,12,13]. Diarrhoeal diseases are listed amongst the top ten causes of death for all sexes and ages in developing countries worldwide [1,9,14,15,16]. Diarrhoeal diseases are listed as the second major cause of death, particularly in developing countries in children under the age of five years resulting from the consumption of microbiologically contaminated drinking water [13,17,18].

Some of the factors that are linked to waterborne disease outbreaks include man-made activities and natural factors [19,20]. These factors may include climate change, water stress, and the inadequate treatment of contaminated groundwater [19,20]. The provision of microbiologically contaminated water to populations serves as a threat to public health and is associated with the spread of infectious diseases, mortalities, and morbidities [10,11,21,22]. Based on recent trends, this study assesses diarrhoea among children under 5 years and microbial drinking water quality compliance in South Africa between the study period 2008–2018.

## 2. Materials and Methods

### 2.1. Study Setting

The study was conducted in South Africa, a country with nine provinces, namely Limpopo, Mpumalanga, North West, Northern Cape, Western Cape, Eastern Cape, Free State, KwaZulu-Natal, and Gauteng. According to Worldometer, South Africa has a population of 61,040,416 with a land area of 1,213,090 KM^2^ [23]. A total of 66.7% of the population of South Africa is urban. This amounts to 39,550,889 people [23]. In total, 6.8% of the population has no access to safe drinking water [23]. Infant mortality has been reported to be 23.6 infant deaths per 1000 live births, and under-5-years deaths are presently reported as 30.7 per 1000 live births [24,25,26]. Approximately 460,250 deaths were caused by water-related diseases worldwide, with about 799,458,315 people having no access to safe drinking water globally; however, in this study the focus was only on South Africa (See Figure 1) as a study setting [24,25,26].

### 2.2. Study Design

The trend analysis was conducted using secondary data from the DWS IRIS database and the DOH DHIS database.

### 2.3. Data Collection

Secondary data for the microbial drinking water quality compliance of all nine provinces in South Africa were sought from the Department of Water and Sanitation (DWS) IRIS database from 2008 to 2018. This database is an integrated regulatory information system that serves as a surveillance system for water quality, where the data for the microbial, chemical, and physical parameters are recorded at the municipal, provincial, and national levels. The data were used to detect the pathogenic microorganisms present in potable water that children under the age of five years in South Africa were exposed to. The recorded concentrations of pathogenic microorganisms found in drinking water were reviewed for all nine provinces in South Africa to check compliance with SANS 241:2015 Edition 2 for drinking water quality in South Africa. Secondary data for the diarrhoeal data for children under the age of five years in South Africa were also requested from the Department of Health DHIS database from 2008 to 2018. The Department of Health District Information system is a database used to keep records of diarrhoeal mortality and morbidity at the district, provincial, and national levels.

### 2.4. Study Population (Inclusion and Exclusion Criteria)

The study population included all children from different races, age groups, and genders under the age of five years in South Africa who consumed portable water and experienced diarrhoea during the period 2008–2018.

### 2.5. Data Analysis

The microbial water quality compliance data and diarrhoeal data for children under the age of five years were merged and compared to determine whether there is a relationship between the concentration of pathogenic microorganisms detected in drinking water and the prevalence of diarrhoea, mortality, and morbidity in children under the age of five years in South Africa. The two datasets for all nine provinces in South Africa were cleaned and merged to assess the trends over time using Microsoft Excel 365 and Stata 14 software. Stata version 14 and Microsoft Excel were employed to observe descriptive statistics, analyse trends, and develop tables and graphs. *E. coli*, *Protozoa*, *Heterotrophic Plate Count*, *Total coliforms*, and *Faecal coliforms* are listed as microbiological indicator organisms for drinking water in South Africa [1,2]. When these parameters were detected in potable water in levels ranging within the stipulated limits in SANS 241:2015 Edition 2, this was considered 100% compliance. An acceptable score for drinking water quality compliance was >95% in an area with a population of 100,000 people and >97% in an area where the population was above 100,000. Compliance status was reported as ‘therperc’ in this study, which indicates the percentage achieved for microbial water quality per pathogen (Figure 2). The pathogenic microorganisms detected in drinking water per province were summarised in tables and graphs using their frequencies and quantities. Data analysis and the interpretation of microbial water quality data and diarrhoeal data for children under the age of five years in South Africa were used to arrive at findings and draw conclusions.

### 2.6. Validity and Reliability

The data consisted of national microbial water quality data for all nine provinces, including both rural and urban areas. The large samples were representative of the target population and allowed for greater validity and the generalisation of findings. The reliability of the data was trusted, given the management of the two databases by two national departments that continuously check the quality of the data.

## 3. Results

The analysis was done in STATA software version 14 using a 95% confidence interval and a 5% significance level, followed by correlational analysis [8].

### 3.1. Descriptive Statistics

The descriptive statistics presented in the form of means and standard deviations, trend analysis of disease, and compliance status, as shown in Table 1, provide an average under 5 years mortality rate per province and its corresponding average compliance status (therperc) for the period 2008 to 2018.

Table 1 indicates Limpopo with the least compliance status of 88.55 for microbial water quality with an aggregate of 21.59 for children under 5 years mortality rates.

Table 1 indicates the highest compliance statuses for microbial water quality in the Western Cape, Gauteng, KwaZulu-Natal, and Northern Cape provinces, with aggregates over 99% and their corresponding under 5 years mortality rates, which were lower with aggregates of 2.15 for Northern Cape and Western Cape, followed by 11.79 and 30.09 for Gauteng and KwaZulu-Natal, respectively (Table 1). Limpopo had the lowest compliance status of 88.6% with an aggregate of 21.59 for under 5 years mortality rates, while Eastern Cape and KwaZulu-Natal had the highest average for under 5 years mortality rates and the highest average compliance statuses for the period 2008 to 2018 (Table 1).

Table 2 indicates the highest average of 738 (SD = 248) new cases of diarrhoea with dehydration between 2008 and 2018 and the lowest average for the child under 5 years diarrhoea case fatality rate, which was 5 per 1000 cases between 2008 and 2018 (Table 2).

### 3.2. Microbial Water Quality Compliance Status Based on SANS 241:2015 Edition 2

The microbiological determinants detected in drinking water should comply with the numerical limits set by SANS 241:2015 Edition 2 to protect and promote human health [1,2]. The analysis below is based on the mean values for *Faecal coliforms*, *E. coli*, *Total coliforms*, and *Heterotrophic Plate Count* over the study years (2008–2018).

Figure 2 shows the aggregated values of water quality indicator organisms detected in drinking water in South Africa for the period 2008 to 2018 at the national level annually. The pathogenic microorganisms that were detected in drinking water include *E. coli*, *Faecal coliforms*, *Total coliforms*, *Heterotrophic Plate Counts*, and *Protozoa parasites.* These pathogens were measured in different quantities. Their compliances ranged from above ninety (90%) to one hundred (100%) for almost all microbiological determinants annually except for *protozoa Giardia* in 2008 and 2009 (Figure 2).

#### 3.2.1. National Trends of Diarrhoea in Children under 5 Years of Age in South Africa

The mortality and morbidity rates for children under the age of five years in South Africa showed a decline (Figure 3). Diarrhoea with dehydration cases decreased immensely over the years, starting from 2008 with 114,811 cases. Ten years later, diarrhoea with dehydration cases declined to 42,296 (Figure 3). Case fatalities were reported to range around 1000 from 2008 to 2018. Diarrhoeal deaths for children under the age of five years in South Africa also indicated a decrease from 2008 to 2018 (Figure 3).

Figure 3 indicates the health outcomes sustained by children under the age of five years in South Africa from 2008 to 2018. Diarrhoea separation, diarrhoea deaths, case fatalities, and diarrhoea with dehydration new cases in children under five years old in South Africa were showcased in different quantities for the duration of 2008 to 2018.

#### 3.2.2. Trends Analysis of *Faecal Coliforms*, *Total Coliforms*, and *E. coli*

##### Trends Per Pathogen Merged with Health Outcomes at the National Level

*Faecal coliforms* are defined as microorganisms that have the ability to grow and have similar fermentative and biochemical properties at 44 °C as they have at 37 °C as defined in SANS: 241:2015 Edition 2 [1,2]. *Faecal coliforms* are listed as indicator organisms for drinking water quality measured in count per 100 mL, and a standard limit has not yet been detected [1,2].

Figure 4 indicates that morbidity decreased from 2008 and reached a record low in 2018. As morbidity dropped in 2008–2011, the compliance for *Faecal* coliforms also dropped. At the beginning of 2012, the compliance levels increased for a year, and morbidity levels dropped. This was followed by a drop in compliance from 2013–2014, a period when morbidity slightly increased. In 2015 compliance increased as morbidity decreased, and in 2016 compliance increased while morbidity dropped. In 2017 compliance levels decreased, and morbidity also declined. In 2018 compliance sharply increased while morbidity seemed to increase slightly (Figure 4).

Figure 4 indicates that under 5 years mortality decreased from 2008 and reached a record low in 2018. Generally, compliance for *Faecal coliform* decreased up until 2015 and increased in 2016 before falling in 2017 and rising again in 2018 (an unsteady trend). As under 5 years mortality decreased from 2008 to 2011, compliance also decreased (a contrary trend), while an increase in compliance in 2012 saw a huge drop in under 5 years mortality. Compliance dropped towards 2013 and in 2014, which resulted in an increase in under 5 years mortality (Figure 5).

*E. coli* are defined as microorganisms that have the ability to ferment lactose with the production of both acid and gas at 44 °C, as defined in SANS 241:2015 Edition 2 [1,2]. *E. coli* is listed as an indicator organism for drinking water quality measured in count per 100 mL, and a standard limit has not yet been detected [1,2].

The under 5 years mortality rate decreased over time with an increase in compliance with pathogens testing. Under 5 years mortality, like morbidity, dropped drastically between 2008 and 2010, which was associated with an exponential increase in the compliance rate. A temporary drop in compliance in 2011 did not affect the under 5 years mortality rate, and compliance increased from 2012 to 2018. This was associated with a drastic decrease in under 5 years mortality (Figure 6).

Figure 7 indicates that *E. coli* pathogen compliance generally increased between 2008 and 2018. There was an exponential increase from 2008 to 2010, which coincided with a drastic drop in morbidity (diarrhoea with dehydration). Compliance dropped slightly in 2011, although this did not affect the decrease in morbidity, and compliance continued to increase until 2018, with morbidity also decreasing until 2018.

*Total coliforms* are defined as a group of aerobic microorganisms and facultatively anaerobic Gram-negative bacteria that have the ability to ferment lactose and are known to be found in the large intestines of humans and animals [1,2]. *Total coliforms* are listed as indicator organisms for drinking water quality measured in count per 100 mL, and the standard limit is ≤10 [1,2].

Compliance with total coliforms increased from 2008 to 2010, which aligned with a decrease in morbidity in 2009 and 2010 (Figure 8). A temporary drop in compliance in 2011 did not affect the decrease in morbidity, and compliance increased from 2012 to 2018. This was associated with a decrease in morbidity.

The under 5 years mortality rate generally decreased with increasing compliance with total coliforms analysis (Figure 9). The diarrhoea case fatality rate followed the same trend as the under 5 years mortality rate, decreasing with increasing compliance. The microbial indicator parameters used in this study for drinking water do not indicate a negative impact on the health of children under the age of five years in South Africa.

### 3.3. Pairwise Correlation

#### 3.3.1. Pearson’s Correlation

Correlation analysis was used to determine if there was a relationship between the occurrence of microbial pathogens in water, morbidity, and mortality rates in children under the age of 5 years. Pearson’s correlation was used to measure the strength and direction of the linear relationship between microbial pathogens in water, morbidity, and mortality rates. The correlation coefficient can range from −1 to +1, with −1 indicating a perfect negative correlation, +1 indicating a perfect positive correlation, and 0 indicating no correlation at all. The *p*-values associated with the correlations were set as 5%, corresponding to a 95% confidence interval, which is the recommended level of certainty [8]. In this study, a negative correlation was found between the occurrence of microbial pathogens in water, morbidity, and mortality rates in children under the age of 5 years in South Africa.

Table 3 indicates that microbial data compliance with SANS 241:2015 Edition 2 was positively associated with diarrhoea death under 5 years (r = 0.236). The diarrhoea case fatality rate was also positively associated with microbial data compliance with SANS 241:2015 Edition 2 (r = 0.226, *p* < 0.05). Diarrhoea with dehydration new cases in children under 5 years and diarrhoea separation for under 5 years did not seem to be associated with microbial water quality compliance over the study period.

#### 3.3.2. Multivariate Regression

Multivariate regression analysis was applied to determine if there was a relationship between the occurrence of microbial pathogens in water, morbidity, and mortality rates in children under the age of 5 years. The dependent variable was under 5 years mortality due to diarrhoea, while the independent variables or predictors were the microbial compliance statuses of drinking water quality, and the province was a control variable. The results are shown in Table 4 below.

Table 4 indicates that there was a negative relationship between the occurrence of microbial pathogens in water (as measured by compliance status) and under 5 years mortality. The results were significant at the 5% level (β = −0.31, *p* < 0.05). These results suggest that as compliance status improved, the mortality rate among children aged under 5 years decreased by 31% over the study period and vice versa. A conclusion can thus be drawn that the microbiological pathogens detected in drinking water at levels complying with SANS 241:2015 Edition 2 drinking water standards and the diarrhoea incidences were not the primary cause for mortality rates in children under 5 years in South Africa between 2008 and 2018.

## 4. Discussion

In general, the microbiological pathogens detected in drinking water in South Africa include *Protozoa species* (*Giardia and Cryptosporidium*), *E. coli*, *Heterotrophic Plate Count*, *Total Coliforms*, and *Faecal coliforms*. The quantities of *E. coli*, *Faecal coliforms*, *Total coliforms*, and *Heterotrophic Plate Count* detected in potable water in South Africa were reported to be compliant with the SANS 241:2015 Edition 2 set limits.

An acceptable score for drinking water quality compliance was considered >95% in an area with a population of 100,000 people and >97% in an area where the population is above 100,000.

Microbial water quality pathogens were measured in different quantities, as indicated in Figure 2, and their compliances ranged from above ninety percent (90%) to one hundred percent (100%) for almost all microbiological determinants annually except for *protozoa Giardia* in 2008 and 2009. *Protozoa Giardia* compliance was low, ranging from below forty percent (40%) in 2008 and towards seventy-four percent (74%) in 2009 (Figure 2). This serves as a public health concern since *protozoa Giardia* is known to cause giardia infection followed by symptoms such as abdominal cramps, watery diarrhoea, nausea, and fatigue [27,28]. *Protozoa Giardia* is also known to be found in developing countries where there is inadequate water supply and sanitation, leading to waterborne disease outbreaks [27,28]. The implementation of SDG 6 and SDG 3 could assist in curbing the spread of diarrhoeal diseases in children under the age of five years resulting from the consumption of microbiologically contaminated drinking water [7].

In 2009, rotavirus vaccines were included in the expanded vaccine programme in South Africa [29]. Rotavirus vaccines are administered as droplets to children under five years of age in South Africa to prevent diarrhoea caused by the *rotavirus* [29]. After the introduction of these vaccines in South Africa, there was a huge reduction in total diarrhoea hospitalisations in children under the age of five years around 2010/11 as compared with 2009 [29,30]. The reductions were noted in both rural and urban areas in all sites [30].

The massive decline over the years in diarrhoeal deaths and case fatalities for children under five years in South Africa could be linked to other interventions, such as the introduction of rotavirus vaccines in the national immunisation programme [31]. It has been reported that rotavirus vaccines have played a huge role in the reduction of diarrhoeal mortality in countries such as the United States of America [31]. After monitoring in African countries, it has further been reported that the introduction of rotavirus vaccines has reduced rotavirus diarrhoea, hospital admissions due to diarrhoea, and severe diarrhoea among children under the age of five years [31]. Therefore, the decrease in child mortality and case fatalities due to diarrhoea in South Africa could be associated with the immunisation of children under five years with rotavirus vaccines [31].

The results suggest that as the compliance status improved, the mortality rate among children under 5 years decreased by 31% over the study period and vice versa. A conclusion can thus be drawn that the microbiological pathogens detected in drinking water at levels complying with SANS 241:2015 Edition 2 drinking water standards and the diarrhoea incidences were not the primary cause of mortality in children under 5 years in South Africa between 2008 and 2018.

Contrary to what other studies have found both internationally and nationally [11,22,32,33], the pathogenic microorganisms content present in drinking water in South Africa ranged within the acceptable and compliant limits stipulated in SANS 241:2015 Edition 2, leading to minimal health risks and health outcomes during the study period 2008 to 2018. Therefore, the high microbial water quality compliance statuses in South Africa were associated with the low occurrences of waterborne diseases and low child mortality and morbidity rates of children under five years in South Africa.

The health effects linked to the consumption of microbiologically contaminated drinking water in children under the age of five years include diarrhoea, dysentery, gastroenteritis, and intestinal infections. Nonetheless, in this study, it was found that the development of diarrhoea with dehydration, diarrhoeal deaths, case fatalities, and diarrhoeal separation for children under the age of five years was not linked to microbial water quality since trends and analysis in South Africa indicate high microbial water quality compliance with the SANS 241:2015 Edition 2 set limits for drinking water quality. The reason for this could be that, though diarrhoea is listed as one of the waterborne diseases affecting children under the age of five years globally, it can be caused by other factors such as unhygienic behaviours, HIV status, malnutrition, and other factors unrelated to microbial water quality [34,35,36,37,38,39,40,41,42]. There is no evidence that proves that the development of health effects amongst children under the age of five years in South Africa was caused by exposure to microbial water quality complying with SANS 241:2015 Edition 2 standards.

## 5. Limitation

The study used secondary data for diarrhoea and microbial water quality from government departments. Secondary data analysis entails the assessment of data that were collected by another party for a different purpose. This serves as a limitation as the researcher must settle for data that were not collected for the particular purpose of the research.

## 6. Conclusions

Based on the findings gleaned from this study, it is concluded that:Microbial water quality complying with the SANS 241:2015 Edition 2 drinking water standards, does not have a negative impact on the health of children under the age of five years in South Africa.The pathogenic microorganisms detected in potable water comply with the SANS 241:2015 Edition 2 stipulated limits.There is no correlation between microbial drinking water and the prevalence of diarrhoea, mortality, and morbidity in children under 5 years in South Africa during the study period.A huge decline in diarrhoeal mortality and morbidity were observed in South Africa over the study period.

## Figures and Tables

**Figure 1 ijerph-20-00598-f001:**
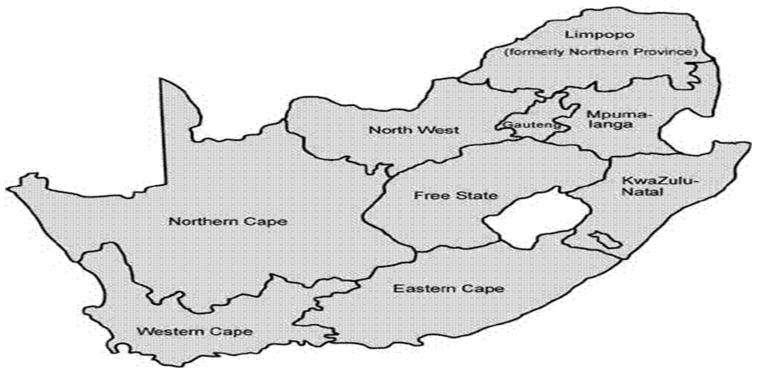
Map of South Africa.

**Figure 2 ijerph-20-00598-f002:**
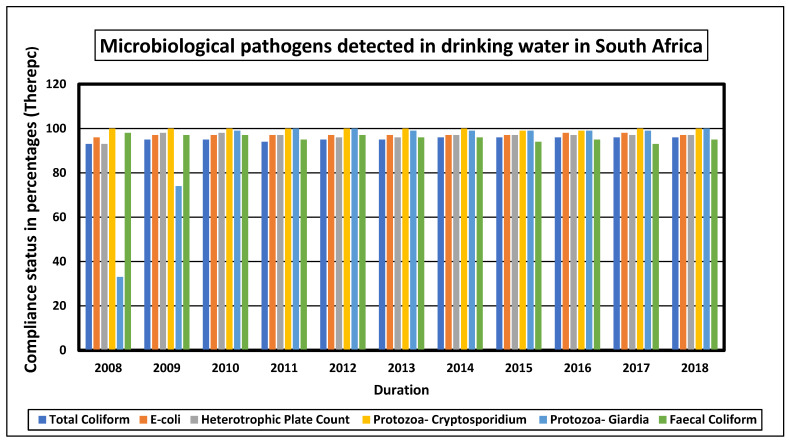
Microbiological pathogens detected in drinking water for all provinces in South Africa annually.

**Figure 3 ijerph-20-00598-f003:**
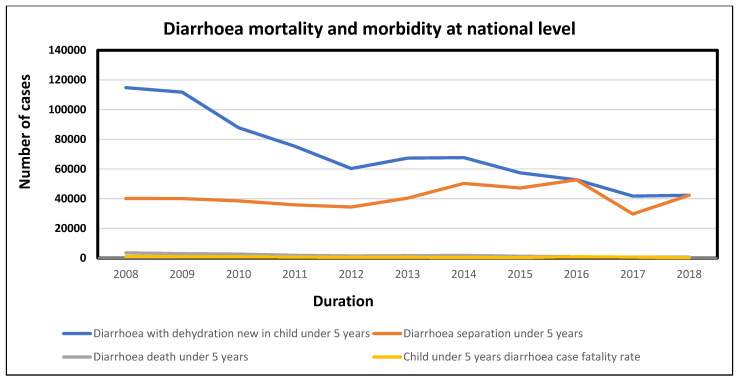
Diarrhoea morbidity and mortality of children under the age of five years.

**Figure 4 ijerph-20-00598-f004:**
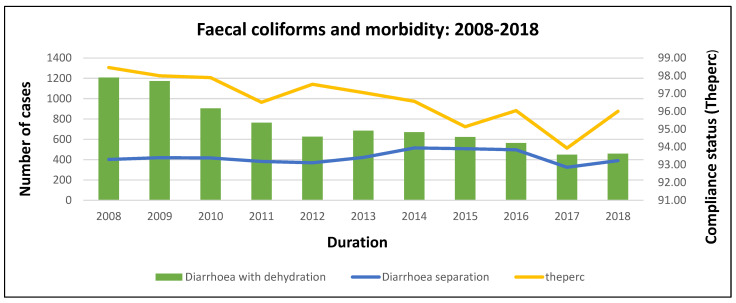
*Faecal coliforms* and morbidity trends at the national level.

**Figure 5 ijerph-20-00598-f005:**
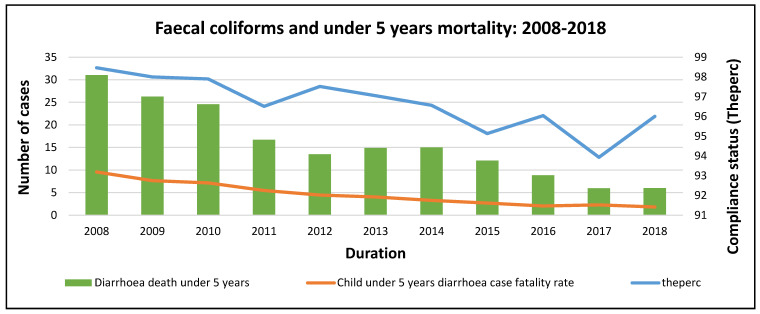
*Faecal coliforms* and mortality trends at the national level.

**Figure 6 ijerph-20-00598-f006:**
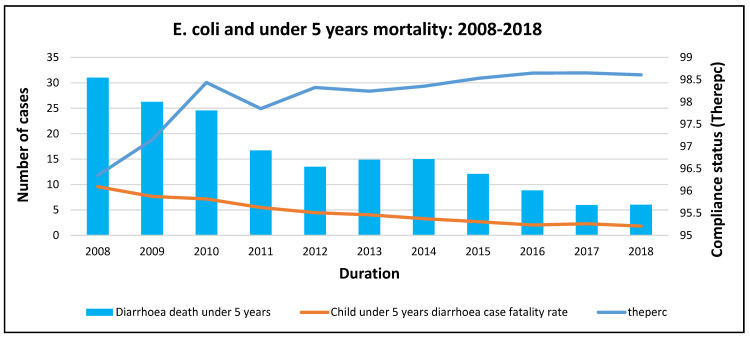
*E. coli* and mortality trends at the national level.

**Figure 7 ijerph-20-00598-f007:**
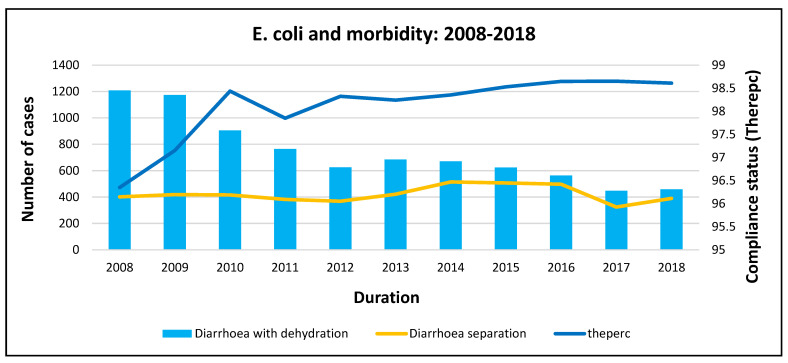
*E. coli* and morbidity trends at the national level.

**Figure 8 ijerph-20-00598-f008:**
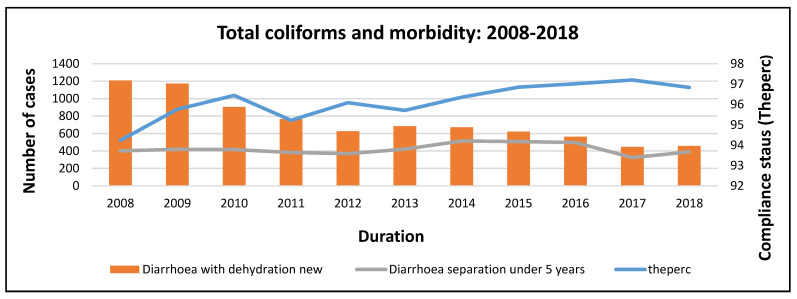
*Total coliform* and morbidity trends at the national level.

**Figure 9 ijerph-20-00598-f009:**
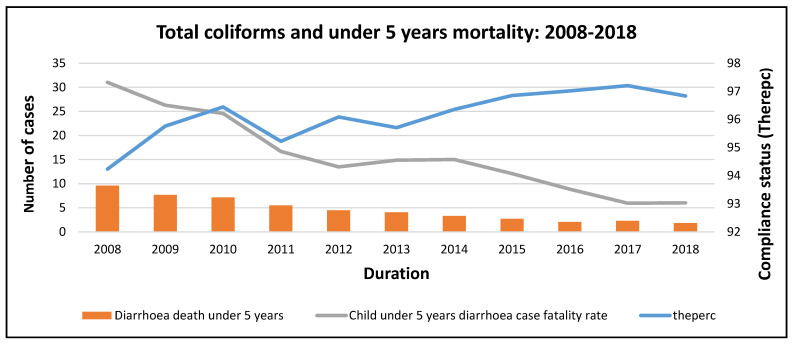
*Total coliform* and mortality trends at national level.

**Table 1 ijerph-20-00598-t001:** Descriptive statistics for children under 5 years mortality rates and their corresponding compliance status (therperc) for per province. (**a**) Limpopo province, (**b**) other provinces.

Province	Under 5 Mortality Rates		Compliance Status (Therperc) at National Level	
	Mean	Std. Dev.	Mean	Std. Dev.
(**a**)				
Limpopo	21.59	11.73	88.55	10.74
(**b**)				
Eastern Cape	34.59	19.00	97.30	1.45
Kwazulu-Natal	30.09	12.53	99.45	0.41
Free State	14.72	14.89	98.65	1.18
Mpumalanga	13.74	7.58	98.44	0.80
North West	12.18	6.87	97.79	0.69
Gauteng	11.79	5.12	99.24	0.26
Northern Cape	2.15	0.64	99.04	0.73
Western Cape	2.15	0.64	99.18	0.52

**Table 2 ijerph-20-00598-t002:** Descriptive statistics for diarrhoea in children under the age of 5 years in South Africa (N = 44).

Variable	Mean	Std. Dev.
Diarrhoea death under 5 Years	15.89	7.96
Diarrhoea with dehydration new cases	738.54	248.79
Diarrhoea separation under 5 years	421.80	58.60
Case fatality	4.590614	2.492427

**Table 3 ijerph-20-00598-t003:** Pearson’s Correlations.

Variable	1	2	3	4	5	6	7
**1. Diarrhoea death under 5 years**	1						
**2. Analysis**	0.298 *	1					
**3. Compliance status**	0.236 *	0.285 *	−0.083	1			
**4. Diarrhoea with dehydration new cases in children under 5 years**	0.464 *	−0.15	0.333 *	0.049	1		
**5. Diarrhoea separation under 5 years**	0.1511	0.198 *	0.417 *	0.105	0.559 *	1	
**6. Diarrhoea case fatality rate**	0.724 *	0.287 *	−0.096	0.226 *	−0.031	0.425 *	1
**7. Province**	0.539 *	−0.168	−0.147	0.042	−0.014	0.201 *	−0.417
**8. Year**	0.519 *	0.399 *	0.380 *	0.111	0.361 *	0.0189	0.597 *

**Table 4 ijerph-20-00598-t004:** Multivariate regression: occurrence of microbial pathogens in water and under 5 years mortality rates.

Diarrhoea Death under 5 years	Coefficient	Std. Err.	t	*p* > t	[95% Conf.	Interval]
**Compliance status**	−0.3119	0.112906	−2.76	0.007	−0.5362	−0.0877
**Diarrhoea separation under 5 years**	**0.00490**	0.00113	4.31	0	0.00264	0.00716
**Diarrhoea with dehydration new cases in children under 5 years**	**0.02896**	0.00341	8.49	0	0.02219	0.03573
**Child under 5 years diarrhoea case fatality rate**	**2.82279**	0.22380	12.61	0	2.37829	3.26729
**Province**	−1.6755	0.24380	−6.87	0	−2.1598	−1.1913
**Year**	0.09245	0.25575	0.36	0.719	−0.4154	0.60040
**_cons**	−160.21	517.002	−0.31	0.757	−1187.0	866.593

## Data Availability

The authors declare that the data that support the findings of this article are available. Diarrhoeal data for under 5 years in South Africa and Microbial data sought from DOH and DWS, can be provided on request.

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
