# Peer review of "Diarrhoea among Children Aged 5 Years and Microbial Drinking Water Quality Compliance: Trends Analysis Study in South Africa (2008–2018)"

_ijerph, 2022, doi:10.3390/ijerph20010598_

Round 1
Reviewer 1 Report
The aim of the study should be more clear formulated. The aim of the study is not what the study is about (see lines 77-79 "(..) and the presence of pathogenic microorganisms exceeding drinking water quality standards/guidelines in South Africa") - in the study (and conclusions) there is about the microbial water quality compliance and the diarrhoea among children under 5 years old. There is one province where the compliance with the quality standard cited (SANS 241:2015, ed. 2) is 88.55, so below the standard (table 3.1) and it's data are not presented separately in results - all the other provinces seem to comply with the cited standard (SANS).
Materials and methods:
line 94 - review how many deaths? 460 or 250?
point 2.4. - study population (lines 115-117) - included all children (...) under the age of 5 who were exposed to microbial drinking water quality pathogens - ALL children under 5 were included, or just those with diarrhoea?
Results:
Information from lines 174-182 - should be placed in the Materials and Methods section.
line 176 - E. coli is abbreviated with a dot, not a line (E-coli)- if you want to change over the paper
Figure 3 - Please, revise the figure - it misses at least the years on the x axis.
lines 213-219 - the information belong better to Discussion section, and it seems that it was added there! (lines 373-379).
line 221 - E. coli - make it italic
Figure 4. - what represents the numbers from left and right? - please, specify (in other graphs, too!). Also, is missing the year 2018 (the number in the figure). Please, verify the explanation of the figure - lines 232 - 238. (ex. the compliance increased in 2016 and not decreased; it decreased in 2017, from the figure). And according to your information (line 151-152), therperc means compliance status. In the figure here and next ones (5-9), is mentioned "theperc".
Figure 5. Inside the figure, in the title, you have an abbreviation (U-5, which is also found in other figures!). Please, explain it or remove the title inside the figure, since you have the title of the figure below.
line 249 - new paragraph when talking about E. coli.
line 278 - the way the name of the figure 8 is edited completely different than the other
What authors mean by "diarrhoea with dehydration new"? (ex. table 3.3 and line 309)
lines 338-340 - need to be deleted
Discussion:
lines 352-358 are repeating! (see lines 176-182) - and should belong to Materials and Methods.
Conclusions:
lines 435-437 - reformulate!
lines 444-445 - to be deleted
Institutional Review Board Statement: to be added in here, too. (line 142).
References:
- to be formulated according to this journal (numbered, numbers in square brackets... see author information).
Author Response
| reviewer 1 | |

Reviewer 2 Report
This study addressed the very important topic about diarrhoea outbreaks. The authors provide the trends analysis of diarrhoea in South Africa during 2008-2018.. However, the manuscript needs some revision before publication. They found that as compliance status improved, the mortality rate among children under 5 years old decreased by 31% over the study period and the microbiological pathogens detected in drinking water at levels complying with SANS 241: 2015 edition 2 and diarrhoea incidences were not the primary cause of the mortality of children under 5 years old in South Africa between 2008 and 2018.
1. Why select the period of 2008-2018?
2. Some figures are directly from statistics software, if they can be modified will be better .
3. Please add more description about DWS IRIS database and the DOH DHIS 100 database.
Author Response
Greetings,
Please see attached responses cover letter.
Thank you so much.
Feida
